# Morphometric Evaluation of Thoracolumbar Spinal Canal and Cord by Magnetic Resonance Imaging in Normal Small-Breed Dogs

**DOI:** 10.3390/ani14071030

**Published:** 2024-03-28

**Authors:** Gabchol Choi, Myungryul Yang, Seungweon Yang, Sungbeen Park, Suyoung Heo, Namsoo Kim

**Affiliations:** 1Jeonbuk Animal Medical Center, Department of Veterinary Surgery, College of Veterinary Medicine, Jeonbuk National University, Iksan-si 54596, Republic of Korea; vcgc@amcw.me (G.C.); jmgohw@naver.com (M.Y.); psb4666@naver.com (S.P.); syheo@jbnu.ac.kr (S.H.); 2Animal Medical Center W, Seoul 04029, Republic of Korea; 3Department of Computer Engineering, College of Science and Technology, Woosuk University, Jincheon-gun 27841, Republic of Korea; swyangy@gmail.com

**Keywords:** intervertebral disc disease (IVDD), magnetic resonance imaging (MRI), spinal cord, spinal canal, age, weight, small-breed dogs

## Abstract

**Simple Summary:**

This study focused on understanding spinal health in small-breed dogs, particularly regarding thoracolumbar intervertebral disc disease, a common spinal injury in dogs. Using MRI, we aimed to establish a baseline for spinal cord and canal measurements across seven intervertebral disc levels (T11–L5). Our hypothesis was that these measurements might vary based on the dog’s body weight and age. The findings revealed that as the body weight increased, the width and height of the spinal cord and canal also increased at all levels. Interestingly, the cord-to-canal ratio of the width showed a negative correlation with the body weight, while the ratio of the height did not. However, there were no significant correlations between these measurements and the age groups. These results provide valuable morphometric reference points for understanding and researching spinal diseases in small-breed dogs, offering insights that could benefit both veterinary care and clinical research.

**Abstract:**

Thoracolumbar intervertebral disc disease (IVDD) is the most common cause of spinal injury in dogs. MRI has been considered the gold standard for neurologic diagnosis, but studies focusing on the thoracolumbar spinal canal and spinal cord using MRI in small-breed dogs are limited. Therefore, this study aimed to establish an MRI reference range for the spinal cord and canal measurements (height, width, cord-to-canal ratio of height, width, cross-sectional area (CSA)) of each intervertebral disc level from T11 to L5 (total of seven levels) on transverse T2-weighted images in normal small-breed dogs. We hypothesized that the spinal cord and spinal canal measurements might vary according to the body weight and age. The width and height of the spinal cord and canal increased as the body weight increased at all levels (*p* < 0.05). The cord-to-canal ratio of the width showed a negative correlation to the body weight at all levels. The cord-to-canal ratio of the height did not show any correlation to the body weight at all levels. All measurements (height, width, cord-to-canal ratio of height, width, CSA) did not show any statistical correlation between the groups subdivided by age. These measurements could serve as a morphometric baseline for thoracolumbar spinal diseases and clinical research in small-breed dogs.

## 1. Introduction

Neuropathy and spinal pain associated with the degenerative disorder, narrowing, or stenosis of the spinal cord or canal is a common feature in small animal practice [1]. Among all spinal diseases, IVDD is the most common cause of spinal injury in dogs [2,3,4]. Thoracolumbar IVDD accounts for 66% to 87% of all IVDD in dogs, and the discs between T12 and L3 have been proven to be at a higher risk of herniation [5].

Imaging of the spine is routinely performed to diagnose such disorders [6]. These disorders can be diagnosed by multiple imaging modalities, such as myelography, computed tomography (CT), and magnetic resonance imaging (MRI) [7,8]. In recent years, due to the advancement of diagnostic techniques, MRI has been considered the gold standard for neurologic problems [9,10,11,12,13].

Quantitative assessment of the spinal cord size in addition to the spinal cord and canal size ratio can provide substantial data for patient prognosis prediction and decision making [14]. Few anatomic and quantitative studies have been published to evaluate the morphometry of normal spinal canal and cord in dogs using MRI and CT [6,14,15,16,17,18]. Most studies have focused on the cervical spinal canal and cord, while some focused on the thoracolumbar spine [6,7,8,9,10,11,12,13,14,19,20,21]. One study used MRI to measure the spinal canal and cord of three specific vertebral body regions (T4, T9, L3) in mid-sagittal images [6]. Another study used CT to measure the C6 and T13 spinal cord-to-canal area ratio in French bulldogs. However, limited resources focus on the thoracolumbar spinal canal and spinal cord using MRI in small-breed dogs.

The purpose of this study was to establish reference ranges for MRI measurements, such as the width, height, and CSA-based cord-to-canal ratio, in the thoracolumbar region, including areas commonly affected by IVDD, in normal small-breed dogs without spinal cord disease. Through this study, we aimed to contribute to establishing a morphometric baseline in normal small-breed dogs and to aid in the understanding and clinical research of thoracolumbar spine diseases in small-breed dogs. We hypothesized that the spinal cord and spinal canal measurements might vary according to the body weight and age.

## 2. Materials and Methods

### 2.1. Sample Collection Criteria

The MRI databases, which included MRI examination of the thoracic and lumbar spine, were collected for our study from four animal medical centers associated with the authors, including the centers with which the authors are affiliated, from January 2017 to January 2022. Also, the following information was collected from the medical records of patients who underwent MRI: signalments, clinical signs, physical examination results (including orthopedic examination and neurologic examination), blood analysis results, and the reasons for MRI. The dogs included in our study were mainly orthopedic patients brought in for a neurologic screening or patients suspected of neurologic disorders, who were not diagnosed as having spinal diseases including spinal deformities such as hemivertebrae. Dogs with a confirmed diagnosis of spinal disease potentially resulting in the increase or decrease in the spinal cord diameter along with a clinical history and symptoms (e.g., intervertebral disc disease, myelitis, or degenerative myelopathy) were excluded from the study. Patients without a physical examination and bloodwork (complete blood cell count, serum biochemistry, and blood gas analysis) data were also excluded.

### 2.2. Measurements and Analysis of MRI

MRI scans were performed under general anesthesia. Midazolam and butorphanol were used as the premedication agents, and propofol was used as the induction agent. Then, general anesthesia was maintained using sevoflurane with 100% oxygen. The patients were all positioned in dorsal recumbency. Sagittal and transverse T2-weighted (T2-W; TR 2100–3100 ms, TE 78–112 ms) images were obtained using a 2T MRI system (GE Healthcare, New York, NY, USA). The slice thickness was 2.5 mm on the sagittal images and 3.0 mm on the transverse images. The images were analyzed with a communication system (INFINITT software version 1.3.2, INFINITT Healthcare, Seoul, Republic of Korea).

Prior to measurements on the transverse images, the thoracic vertebra (T10) to lumbar vertebra (L5) were inspected for intervertebral disc herniation on the midsagittal T2-W images. Measurements on the transverse images followed the previously described methods [11,16,17,19]. On the transverse T2-W images, the height and width of the spinal canal were measured (Figure 1A). The height of the spinal canal was drawn at the level of the bisecting line of the vertebra. The width of the spinal canal was then drawn perpendicular to the height line. The height and the width of the spinal cord were measured using the same steps as described above (Figure 1B). In addition to the height and width measurements of the spinal cord and canal, the cord-to-canal ratio according to the width, height, and CSA was calculated. Assuming the ellipse shape of the spinal canal and cord, the cord-to-canal ratio of the CSA was calculated as the product of the spinal cord height and width divided by the product of the spinal canal height and width (Figure 1C). From intervertebral disc space T11–T12 to L4–L5, the spinal cord and canal measurements of all intervertebral disc spaces were evaluated sequentially.

The evaluation was performed by a single veterinary radiologist specialized in MRI interpretation. The mean of these measurements was calculated from the measurement of each sample in triplicate.

### 2.3. Sample Population and Group

Two hundred and twenty-five client-owned dogs were included. The animals were divided into two groups according to their body weight and age. According to the body weight, the animals were subdivided into three groups (<5 kg, ≤5–<10 kg, and ≤10–<20 kg). According to their age, the animals were divided into three groups (≤5 years, <5–<10 years, and ≤10 years).

### 2.4. Statistical Analysis

Statistical analyses were performed using GraphPad Software (GraphPad Prism 9.0, Boston, MA, USA). Data are presented as the mean ± SD. Homogeneity of variance was assessed using Levene’s Test for equality. A one-way ANOVA was performed to analyze the effect of the body weight and age on each spinal cord and spinal canal measurement. The Scheffe post hoc test was performed to correct for multiple comparisons. Welch’s ANOVA followed by the Games–Howell post hoc test were conducted when unequal variances were detected. *p* value < 0.05 was considered statistically significant.

## 3. Results

In total, 225 dogs including 22 different small breeds were selected from hospital records for this study. The breeds in this study included: Maltese (52), Mixed Breed (40), Dachshund (32), Toy Poodle (19), Pomeranian (18), French Bulldog (12), Pekingese (11), Coker Spaniel (8), Shih Tzu (7), Bichon Frise (5), Miniature Pinscher (3), Welsh Corgi (3), Pug (2), Beagle (2), Yorkshire Terrier (2), Chihuahua (2), Spitz (2), Schnauzer (1), Bulldog (1), Papillon (1), King Charles Spaniel (1), and Coton de Tulear (1). There were 123 male dogs and 102 female dogs in this study. The mean body weight was 6.58 kg (range, 1.1–20 kg), and the mean age was 7.77 years (range, 1–14 years).

The measurements and ratio for each disc level (seven levels; T11–T12 to L4–L5) subdivided by body weight and age are summarized in the Figures. The measurements included the height and width of the spinal cord and spinal canal, as well as the cord-to-canal ratio of the height and width.

The height and width of the spinal cord and canal increased as the body weight increased at all levels (*p* < 0.05) (Figure 2). The cord-to-canal ratio of the height in the body weight group did not show any positive or negative correlation at all disc levels, and there were data with large SD values compared to the other measurements. As a result, statistical significance was not confirmed (Figure 3). The cord-to-canal ratio of the width showed a negative correlation to the body weight in all levels, but no statistical significance was found in the T11–T12 and L4–L5 levels (Figure 3).

The cord-to-canal CSA ratio was between 0.52 and 0.61 at all disc levels subdivided by body weight. The largest was at the level L4–L5 in dogs under 5 kg (0.61), and the smallest was at T13–L1 in dogs between ≤10 kg and <20 kg (0.52) (Figure 4). The cord-to-canal CSA ratio was between 0.53 and 0.60 at all disc levels subdivided by age. The largest was at the level L3–4 in dogs ≤ 5 y (0.60), and the smallest was at the level T11–T12 in dogs ≥ 10 y (0.53) (Figure 4).

Between groups subdivided by age, the spinal cord and canal measurements and the cord-to-canal ratio did not show any statistical correlation (Figure 5).

## 4. Discussion

MRI has become the gold standard technique for diagnosing spinal diseases in veterinary practice. The thoracolumbar region accounts for the largest proportion of spinal diseases, such as IVDD amongst all regions. Surgical decompression of the spinal cord is one of the main treatment options for IVDD. Minimally invasive surgery of the thoracolumbar region has attracted attention in veterinary surgery [19,21,22,23,24,25]. A surgical approach assisted with an endoscope, arthroscope, or microscope aims to minimize iatrogenic damage to the surrounding tissue, which enhances healing and reduces complications. Due to this technical nature, the surgical field of view around the spinal structures is limited, which makes the morphometric measurements of the thoracolumbar spine more important. To our knowledge, there are limited reports focused on normal MRI morphology of the thoracolumbar spine in small-breed dogs [6].

In this study, the width and height of the spinal cord and canal showed a tendency to increase as the body weight increased. The spinal cord-to-canal ratio of the width showed negative correlations to the body weight, whereas no correlations were found between the body weight and the cord-to-canal ratio of the height. There was no significant correlation between the age and the spinal cord and canal measurements at all levels. Therefore, these findings suggest that the width, height of the spinal cord and canal, and the spinal cord-to-canal ratio of the width are dependent on body weight rather than on age in small dogs.

Over the past two decades in human spinal surgery, endoscopic surgery has made significant technical progress, and the vast majority of spinal procedures can be performed endoscopically [26,27,28]. Intralaminar and transforaminal approaches are mainly used, and the introduction of the endoscope and surgical instruments into the spinal canal is inevitable. The cord-to-canal CSA ratio of the thoracolumbar regions in normal small-breed dogs could be useful for further minimally invasive endoscopic spinal procedures in veterinary surgery. The cord-to-canal CSA ratio was between 0.52 and 0.61 at all disc levels. The largest was at the level L4–L5 in dogs under 5 kg amongst all groups (0.61), and the smallest was at T13–L1 in dogs between ≤10 kg and <20 kg (0.52). The CSA ratio tended to increase as we moved to the posterior disc level regardless of the body weight or age. Since our study focused on evaluating each disc level based on the body weight and age differences, we did not evaluate the statistical differences in the CSA ratio between disc levels. Further studies focused on comparing the cord-to-canal area differences between each disc level in the thoracolumbar region are recommended.

According to a previous study, thoracolumbar IVDD is known to show crucial neurologic signs compared to cervical IVDD [21]. The study found that the predominance of neurologic deficits in the thoracolumbar regions has been assumed to be caused by the larger spinal cord-to-canal area ratio of the thoracolumbar region to the cervical region [21]. However, Sara et al., 2022 measured the spinal cord-to-canal area ratio at the mid-body of C5, L1 level in French bulldogs with CT and revealed that the thoracolumbar region had a relatively smaller spinal cord-to-canal ratio compared to the cervical region [18]. Also, in this MRI study, the cord-to-canal CSA ratio was between 0.524 and 0.56 in the T13–L1 and L1–L2 area, which was smaller than the cord-to-canal CSA ratio of 0.58–0.60 in the C5, C5-C6, and C6 level of the previous MRI morphometric study on normal small dogs [17]. These results follow the suggestion that the epidural area, which means the area within the spinal canal that excludes the spinal cord, is not smaller in the thoracolumbar region than in the cervical region. However, further research focused on the comparison between the cervical and thoracolumbar regions in small dogs should be made to confirm this on MRI.

Hecht et al. (2014) reported MRI measurements at the mid-vertebral body level of T4, T9, and L3 in sagittal images. They found no significant correlations between the spinal cord height and body weight, but they found a significant positive correlation between the spinal canal height and body weight [6]. In the present study, no statistically significant differences were found between the spinal cord height and the body weight nor the spinal canal height and the body weight in all the middle-of-the-disc levels through T11–L5. There is a partial disagreement with the previous study. The reasons for this inconsistency could be due to the different sizes and breeds of the dogs, different section levels of measurements (mid-vertebral body or middle of the disc), and different measurement planes (sagittal or transverse).

Regardless of the body weight or age, the height and width of the spinal cord and canal at levels L3–L4 and L4–L5 were significantly larger than at other levels. The lumbar enlargement, which is known to give rise to the lumbosacral plexus, lies at the fourth and fifth vertebrae [29]. This natural enlargement could have been reflected in our findings.

The spinal canal of dogs is relatively oval compared to humans [30]. Based on this, the spinal cord-to-canal CSA ratio was calculated as the product of the spinal cord height and width divided by the product of the spinal canal height and width, assuming the ellipse shape of the spinal canal and cord. This differs from the manual tracing methods of the CSA area, which have been applied in other studies [17]. The reason for choosing this new method was the elliptical shape of the canine spinal cord compared to humans and the expectation that it would result in fewer errors compared to the previous method. However, due to the differing methodology employed in this study compared to previous studies, caution is advised when interpreting the spinal cord-to-canal CSA ratio of this study.

Due to the retrospective nature, some neurologic screening tests, such as cerebrospinal fluid analysis were not carried out in all cases. Some patients with spinal disorders could have been included. Also, the grouping of dogs was based only on body weight and age, without consideration of the body condition or breed disposition of the patients. The data were collected from four different animal medical centers, which could have caused differentiation when positioning the patients for MRI acquisition. Additionally, measurements were conducted by a single observer. Despite performing triplicate measurements for each MRI image, there remains the potential for observer bias to influence the outcomes. Nonetheless, despite these limitations, the study’s extensive sample size has enabled the mitigation of certain technical and subjective errors, thereby enhancing the overall reliability.

## 5. Conclusions

In conclusion, this study provides the height and width of the spinal cord and canal measurements, as the cord-to-canal ratio according to the width, height, and CSA of the thoracolumbar region in normal small-breed dogs. We found a significant increase in the height and width of the spinal cord and canal and a significant decrease in the cord-to-canal ratio according to the width with increasing body weight. There was no correlation between the measurements and the age. These measurements could serve as a morphometric baseline for thoracolumbar spinal diseases and clinical research in small-breed dogs.

## Figures and Tables

**Figure 1 animals-14-01030-f001:**
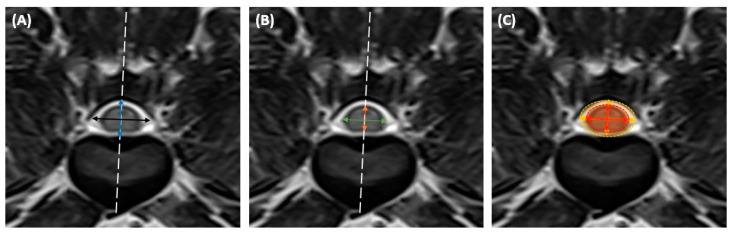
Transverse T2-weighted MR images at the disc level L3–L4: (**A**) The height and the width of the spinal canal. The height of the spinal canal (blue arrow) was drawn at the level of the bisecting line of the vertebra (dashed line). The width of the spinal canal (black arrow) was drawn perpendicular to the height line. (**B**) The height and the width of the spinal cord. The height (orange arrow) and the width (green arrow) of the spinal cord were drawn using the same method. (**C**) The CSA of the spinal cord and canal. The CSA of the spinal canal (yellow area) and the cord (red area) were calculated as the product of the major and minor axes of the ellipse and the circumference, assuming the cross section was elliptical.

**Figure 2 animals-14-01030-f002:**
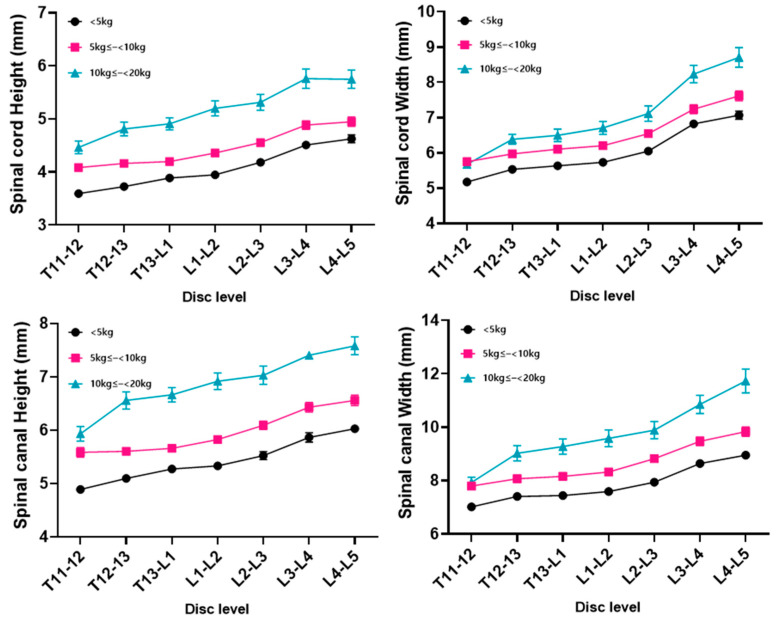
The height and width of the spinal cord and spinal canal by disc level for each body weight group. The dimensions of both the spinal cord and spinal canal demonstrated a consistent increase in height and width with a higher body weight across all disc levels, revealing a statistically significant correlation (*p* < 0.05).

**Figure 3 animals-14-01030-f003:**
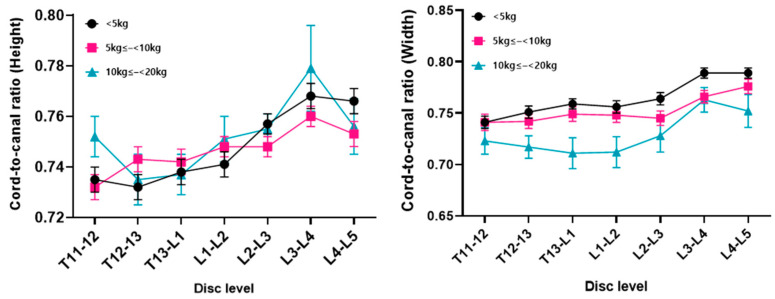
The cord-to-canal ratio of the height and width by disc level for each body weight group. No correlation was observed between the cord-to-canal ratio of the height and the body weight across all levels, and there was no statistical significance. Conversely, the cord-to-canal ratio of the width exhibited a significant negative correlation with the body weight at all levels (*p* < 0.05), except for the T11–T12 and L4–L5 levels, where no statistical significance was identified.

**Figure 4 animals-14-01030-f004:**
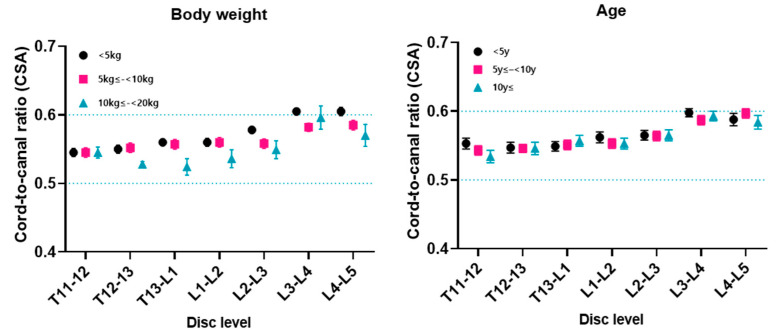
The cord-to-canal CSA ratio by disc level for each weight and age group. The cord-to-canal cross-sectional area (CSA) ratio ranged from 0.52 to 0.61 across all disc levels, stratified by body weight. Notably, the highest ratio was observed at the L4–L5 level in dogs weighing less than 5 kg (0.61), while the lowest was recorded at the T13–L1 level in dogs within the 10 kg to <20 kg weight range (0.52). When categorized by age, the cord-to-canal CSA ratio varied between 0.53 and 0.60 across all disc levels. The maximum ratio was noted at the L3–L4 level in dogs aged 5 years or younger (0.60), whereas the minimum was found at the T11–T12 level in dogs aged 10 years or older (0.53).

**Figure 5 animals-14-01030-f005:**
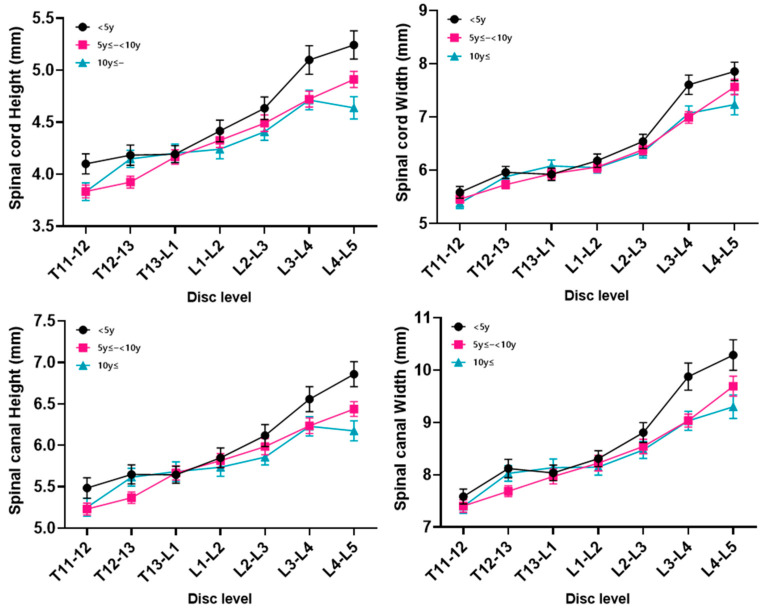
The height and width of the spinal cord and spinal canal by disc level for each age group. Among the age-stratified groups, there was no statistically significant correlation observed in the measurements of the spinal cord and canal, as well as the cord-to-canal ratio.

## Data Availability

The data presented in this study are available on request from the corresponding author upon reasonable request.

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
