# Peer review of "Morphometric Evaluation of Thoracolumbar Spinal Canal and Cord by Magnetic Resonance Imaging in Normal Small-Breed Dogs"

_animals, 2024, doi:10.3390/ani14071030_

Round 1

Reviewer 1 Report

Comments and Suggestions for Authors

I express my congratulations to the authors for their article.The title piqued my interest, leading me to accept the revision.

However, the abstract fails to clearly articulate the study's objective. Unfortunately, even after reading the entire article, the purpose remains unclear. What is the aim? Is it to establish a height and/or width range for determining something specific? In cases of IVDD or other alterations, relying solely on a range may not be sufficient, especially when considering other aspects such as the interruption and/or deviation of the columns of cerebrospinal fluid in T2W pulse sequences, loss of the hyperintense signal in epidural fat, changes in the shape of the spinal cord, and alterations in the normal ovoid shape of the disc, which are best appreciated on transverse images.

Moreover, there are already studies in the literature that describe spinal cord compression ratios. It is essential for the authors to provide a clear and thorough explanation of the aim of their work.

Moving on to the materials and methods, it is crucial to state from the beginning that the study involved four centers. The multicenter nature of the work is a limitation that should be acknowledged. The exclusion criteria need to be defined more precisely, and it is advisable to exclude patients with conditions such as hemivertebrae.

The statistical analysis has been executed accurately; tables and figures are commendable.

The discussion section contains numerous sentences without references, and there is a reference to the cervical spine, which appears inconsistent with the apparent focus on the thoracolumbar tract.

Specific points warrant attention:

Line 53-54: please add the references when you wrote about the study on cervical spine and thoracolumbar one.

Line 59: why a range? There are other signs that allow neurologist and radiologist to assess alteration in the spine width/ height.

Line 65: authors should specify which information was recorded from each patient.

Line 67: why did the authors include the cervical tract if the study is on the thoracolumbar tract?

Line 69: what does “orthopedic patients  brought for a neurologic screening “ mean? If MRI scans were not performed for the study of the spine, these would not be useful for evaluation.

Line 180: this sentence lacks of references.

Line 183: It is still unclear to me why morphometric measurements (and having a range) serve the purpose of surgery. I think it should be explained why.

Line 208: add references.

Line 219: the study is on the thoracolumbar tract, why consider the cervical tract?

Line 236: add references

Line 243: four different center: I reckon authors should better specify the materials and methods of this study.

Line 246: the authors have not taken into consideration the fact that some breeds among those enrolled very often present congenital alterations of the vertebrae (I see French pugs and bull dogs enrolled), I believe that for the purposes of a morpholgical evaluation extreme cases should be considered separately. In addition, the presence of congenital alterations (hemivertebrae, supernumerary vertebrae, etc.) should be added as an exclusion criterion.

Author Response

Thank you so much for taking the time to review our manuscript.

Here are the answers to the points you mentioned.

Thanks again for your help.

Reviewer 2 Report

Comments and Suggestions for Authors

Dear Authors,

Congratulations on your work so far. Here are some of the concerns I have about your study:

1. You have considered these measurements, but what is your opinion on age? Can it be an essential factor in changes occurring in the spinal canal? Especially as you have quite a wide variety included in the study.

2. How does your study influence the diagnosis and treatment of IVDD in small dogs?

Comments on the Quality of English Language

No comment.

Author Response

(The authors gave the same response as above.)

Reviewer 3 Report

Comments and Suggestions for Authors

The aim of this study is to describe the variation in the size of the spinal canal in the thoracolumbar vertebrae and the variation that exists in relation to different weights and ages of the animals. 

It is concluded that there is no variation between age and the measurements taken, however there is an increase in the height and width of the spinal canal as body weight increases and a decrease in the spinal cord:spinal canal ratio as weight increases. 

The strength of this article lies in the number of patients studied (N=255) so the results are very conclusive and with few difficulties in presenting deviations or biased. 

On the other hand, it lays the foundations for the introduction or not of less invasive techniques in the treatment of intervertebral disc disease.

The statistical results, to the best of my knowledge, appear to be correct.

Specific comments:

In materials and methods you should describe the anaesthetic protocol, in case any of the drugs used have an effect on cerebrospinal fluid production, although this is unlikely.

It should explain the variation shown in the cord-to-canal ratio (height) figure. The variability shown in the graph does not represent the results expressed or descriptive information is missing.

Since animals have been used, it should be reflected that permission has been obtained from the owners to use their animals for the indicated imaging procedures, even if for purposes other than the study.

Too many references are made to minimally invasive surgical techniques and the usefulness of this measurement for their use. I think it is used to justify this study too much, because it is not a technique that has yet been proven useful and practical in veterinary medicine, as it has been in human medicine. More scientific evidence is needed and this article does not provide sufficient information on the surgical technique, so it should focus only on anatomical description and advanced MRI imaging.

Please review the discussion and limit the information on minimally invasive techniques in veterinary spinal surgery to the minimum possible. 

Author Response

(The authors gave the same response as above.)

Reviewer 4 Report

Comments and Suggestions for Authors
The manuscript submitted by the authors provides interesting information about measurements and correlations of the spinal canal and cord in small breed dogs. Nonetheless, there are some concerns mainly focused on the discussion section that should be addressed before acceptance.   Specific comments    Line 55, please specify with MRI or CT. Lines 114-115, are you sure that are four groups? Lines 133-134, please add again the information levels. Line 246, please explain better. In discussion, line 175, add gold standard technique. Line 185 of discussion, please add references. Lines 210-218, it is a little confuse, please go deeper or clarify. Lines 220-223, same, please clarify. Line 238, then, here you should explain why you decide to do these measurements instead  those ones. Moreover, you should add references. Lines 239-248, these are the study limitations mainly due to the retrospective nature of the study. Then, put them all together.

Author Response

(The authors gave the same response as above.)

Round 2

Reviewer 1 Report

Comments and Suggestions for Authors

I thank the authors for considering my comments regarding their article. I believe it is now suitable for publication. 

The introduction is comprehensive and accompanied by up-to-date references. The materials and methods are clearly articulated, presenting all factors in a lucid manner. The results, supplemented with suitable tables and figures, facilitate clear comprehension. The discussion exhaustively elucidates the insights garnered from the conducted work.

Reviewer 2 Report

Comments and Suggestions for Authors

Nothing to add.

Comments on the Quality of English Language

Minor editing of English is required.

Reviewer 4 Report

Comments and Suggestions for Authors

The revised version of the paper has addressed all the concerns and can be accepted in its present form.